# `Bileve`: Securing Text Provenance in Large Language Models Against Spoofing with Bi-level Signature

**Tong Zhou**
Northeastern University, Boston
zhou.tong1@northeastern.edu

**Xuandong Zhao**
UC Berkeley
xuandongzhao@berkeley.edu

**Xiaolin Xu**
Northeastern University, Boston
x.xu@northeastern.edu

**Shaolei Ren**
UC Riverside
shaolei@ucr.edu

## Abstract

Text watermarks for large language models (LLMs) have been commonly used to identify the origins of machine-generated content, which is promising for assessing liability when combating deepfake or harmful content. While existing watermarking techniques typically prioritize robustness against removal attacks, unfortunately, they are vulnerable to spoofing attacks: malicious actors can subtly alter the meanings of LLM-generated responses or even forge harmful content, potentially leading to the wrongful attribution of blame to the LLM developer. To overcome this, we introduce a bi-level signature scheme, `Bileve`, which embeds fine-grained signature bits for integrity checks (mitigating spoofing attacks) as well as a coarse-grained signal to trace text sources when the signature is invalid (enhancing detectability) via a novel rank-based sampling strategy. Compared to conventional watermark detectors that only output binary results, `Bileve` can differentiate 5 scenarios during detection, reliably tracing text provenance and regulating LLMs. The experiments conducted on OPT-1.3B and LLaMA-7B demonstrate the effectiveness of `Bileve` in defeating spoofing attacks with enhanced detectability. Code is available at https://github.com/Tongzhou0101/Bileve-official.

## 1 Introduction

Watermarks have been envisioned as a promising method to differentiate content generated by large language models (LLMs) from human [5, 11, 23, 13, 1, 4]. It involves injecting statistical signals into the token sampling process utilizing a secret watermark key. Subsequently, the one who knows the key can verify the content's origin by assessing the presence of the predefined signal through a statistical test. Current watermarking schemes primarily focus on user-side concerns, striving to achieve robustness against watermark removal attacks (i.e., perturb the generated text to remove the watermark), thereby combatting academic dishonesty and other deceptive practices.

However, a critical vulnerability remains unaddressed in watermark design: *spoofing attacks directed towards model owners*. In these attacks, malicious actors attempt to falsely attribute content generated by humans or other models to the targeted model, with the aim of evading accountability[1] or damaging the model's reputation. A few recent works have identified two kinds of spoofing attacks targeting the LLM watermark by exploiting either its symmetric characteristic [6, 15] or learnability [8, 10], as detailed in Sec. 2.3. Furthermore, we propose a new spoofing attack, named semantic manipulation,

---

[1]Tesla lawyers claimed that Elon Musk's past statements about self-driving safety could be deepfakes, which is found suspicious by the court [7].

Table 1: Comparison of different watermarking methods based on four desired properties.

| Methods | Robust | Unforgeable | Tamper-evident | Transparent |
|---|---|---|---|---|
| Kirchenbauer *et al.* [11] | ✓ | ✗ | ✗ | ✗ |
| Zhao *et al.* [23] | ✓ | ✗ | ✗ | ✗ |
| Kuditipudi *et al.* [13] | ✓ | ✗ | ✗ | ✗ |
| Liu *et al.* [15] | ✓ | ✗ | ✗ | ✗ |
| Fairoze *et al.* [6] | ✗ | ✓ | ✓ | ✓ |

which enables attackers to alter the sentiment of generated content with minimal token modifications, as described in Sec. 3.2. It assumes the most constrained capabilities of attackers, where they only have access to the victim model's detector. Due to the robustness of LLM watermarks against perturbations, this attack can manipulate the originally helpful content into something harmful or offensive without compromising the detectability of the watermark, thus successfully achieving spoofing attacks.

Given the serious consequences of spoofing attacks, it is highly in demand to answer the question: ***How to avoid an LLM being wrongly blamed?*** To solve this problem, we aim to design a watermark for LLM, which focuses more on the model owners' side instead of only watermarking on the users' side. To reliably identify the provenance of machine-generated content while being able to defend against the above spoofing attacks, a signature should have the following properties:

- **Robust:** The signature remains capable of tracing the source of machine-generated text even when subjected to certain perturbations, ensuring it is not overly fragile or easily rendered ineffective.
- **Unforgeable:** The signature is inherently resistant to being learned given the components utilized in its detection.
- **Tamper-evident:** It should be able to check the integrity of the generated content, showing reliable tampering evidence to safeguard the interests of model owners.
- **Transparent:** It is detectable without needing access to generation secrets or relying on a black-box API, allowing independent, reliable verification.

Despite the critical importance, achieving all desired properties in a single LLM watermark remains challenging, as even state-of-the-art (SOTA) designs cannot meet them all (see Tab. 1). Indeed, designing such a watermarking scheme involves a fundamental trade-off between defending against removal attacks and spoofing attacks. Specifically, being robust to removal attacks requires that the watermark's detectability remains unaffected by certain perturbations, while anti-spoofing demands sensitivity to perturbations to verify text integrity, distinguishing harmful content from genuine model output and tampered content.

To overcome the above challenges, we propose `Bileve`, a novel sampling strategy by embedding a bi-level signature into generated tokens. At the coarse-grained level, we utilize statistical signals across the entire text to detect the presence of the watermark, ensuring robustness against perturbations. Concurrently, at the fine-grained level, we integrate content-dependent signature bits into each token to uphold content integrity, which leverages a digital signature scheme to ensure unforgeability, as the secret key required for watermark embedding will be securely held by model owners. This scheme enables transparent detection by allowing verification with a public key instead of embedding secrets, so independent parties can authenticate without proprietary details or a black-box API, ensuring reliable detection. And the tampering evidence will show when these two level detection results are not consistent.

Our contributions are threefold: 1) We uncover an advanced spoofing attack that exploits the robustness of SOTA watermarking schemes; 2) We introduce `Bileve`, the first watermarking scheme to simultaneously ensure robustness and unforgeability by embedding a bi-level signature through a novel rank-based sampling strategy; 3) `Bileve` is capable of distinguishing five distinct scenarios during the detection phase, effectively defeating spoofing attacks and serving as a promising tool to regulate LLM safety mechanism.

## 2 Background and Related Works

### 2.1 Language Model Basics

Let $\mathcal{M}$ denote a language model with a vocabulary $\mathcal{V}$ containing $K := |\mathcal{V}|$ tokens. To generate the next token $w_t$, $\mathcal{M}$ will take prior tokens $w_{1:t-1}$ as the input and output a vector of logits $l^{(t)}$,

Table 2: Three categories of spoofing attacks (ranked by attackers' capabilities from high to low).

| Methods | Exploited Vulnerabilities | Attackers' Capabilities |
|---------|---------------------------|-------------------------|
| [6, 15] | Symmetry | Know the secret key for embedding watermarks |
| [8, 10] | Learnability | Get access to the victim model and query it multiple times |
| Our attack | Robustness | Only get access to the victim models' detector |

which is transformed into a probability distribution $\mathcal{D}^{(t)} = (p_1^{(t)}, ..., p_K^{(t)})$ via the softmax function. Then the sampling strategy is applied to determine how the model selects $w_t$ based on $\mathcal{D}^{(t)}$. One common sampling strategy is multinomial sampling, where $\mathcal{M}$ randomly selects the next token from $\mathcal{V}$ according to the probabilities $p_k^{(t)}$ assigned to each token. This process is repeated iteratively to generate a sequence of tokens.

## 2.2 LLM Watermarks

Watermarks for model-generated texts are used to identify the provenance of the text, ensuring accountability in cases where generated content needs to be traced back to a specific LLM. The existing watermark schemes rely on the specialized decoding algorithm to embed statistical signals into generated contents, then enabling watermark detection via computing p-value [1, 11, 23, 4, 13]. For instance, for generating the next token, one approach dynamically partitions the vocabulary into green and red lists based on its previous few tokens and a watermark key [11], then increasing the logits of green tokens to enhance their chance of being selected. During detection, the watermark detection key is used to count the number of green tokens in the text, with the calculated z-statistic indicating the existence of the watermark. Moreover, Zhao *et al.* [23] simplify the scheme proposed in [11] by fixing the green-red list for each token, demonstrating that their watermark is twice as robust to edit as [11]. Furthermore, unlike modifying logits, a distortion-free watermark is proposed to preserve the original text distribution [13]. It leverages robust sequence alignment to align watermarked text to a watermark key sequence in the sampling phase, e.g., using exponential minimum sampling.

However, these watermarking schemes only enable detection by individuals possessing the key, which doesn't facilitate transparent regulation. On one hand, making the key public is susceptible to attacks [15]. On the other hand, maintaining detection privately (e.g., via APIs) compromises reliability, as it functions as a black box, allowing the model owner to manipulate detection results.

## 2.3 Spoofing Attacks

Spoofing attacks can fall into three categories based on the capability of attackers, and each of them exploits different vulnerabilities in SOTA watermarks, as summarized in Tab. 2. First, due to the watermark embedding and detection process sharing the same secret key (i.e., symmetric schemes), the semi-honest detector knowing the secret key can embed the watermark to any content. Such a vulnerability has been discussed in [6, 15], where they design asymmetric watermarking schemes so that detection does not rely on the key used for embedding. Specifically, [6] proposes embedding watermarks using digital signature algorithms, ensuring that only model owners possess the secret key for watermark embedding while providing detectors with access to the public key for detection. However, it is easily broken once the message tokens are perturbed. Besides, it suggests using error-correcting encoding to improve robustness, which unfortunately would increase the risk of spoofing attacks and should not be adopted. Moreover, [15] employs two distinct neural networks for watermark embedding and detection, leveraging an asymmetric scheme for public detection.

However, [15] is based on [11] and thus can be learned as studied in [8], which is the second kind of spoofing attacks. In particular, [8] demonstrated that, by querying the victim model and collecting its watermarked samples, attackers can fine-tune an adversary model utilizing a sampling-based watermark distillation technique to learn the watermark (detailed in Appendix A). The fine-tuned adversary model can respond to any malicious requests, with the response containing the watermark of the victim model. Recently, an independent work also proposed attacks by exploiting watermark robustness [18]. Beyond techniques such as randomly inserting toxic tokens or modifying tokens to alter sentence accuracy, our attack leverages a reward model to guide targeted semantic manipulation. More importantly, while they suggest compromising watermark robustness to mitigate spoofing attacks, our work demonstrates how to achieve effective mitigation without sacrificing robustness.

# 3 Potential Attack: Semantic Manipulation

## 3.1 Threat Model

**Attackers' Objective.** Given text generated by the victim LLM, attackers seek to alter the semantic meaning of the text with minimal changes, transitioning it from something helpful or neutral to harmful or offensive. Owing to the robustness of the existing watermarks, the watermark detector can still identify the presence of the watermark in the modified content. Consequently, the altered content erroneously attributes its origin to the victim model, potentially damaging the model's reputation.

**Attackers' Capability.** Contrary to existing spoofing attacks, our approach assumes the strictest attacker capabilities, where the adversary only gains access to the watermark detector, as outlined in Tab. 2. These attackers lack knowledge of the secret key and are not required to query the victim LLM multiple times to acquire watermarked samples for training other adversary models. They may utilize public language models to enhance the efficiency of their attacks.

## 3.2 Attack Method

We identify the dilemma of being robust and unforgeable. In particular, being robust indicates that the watermark should be preserved after perturbation, thus we can exploit this characteristic to design a novel spoofing attack. Specifically, attackers can query the victim model with harmless prompts, and then use basic word replacement techniques to change its semantic meaning to be toxic or harmful. *Due to the robustness properties of LLM watermarks, the detectability will not be compromised if the portion of word replacement is low.* Consequently, a detector cannot discern whether the content originated from the victim model or was manipulated by malicious actors. This highlights the limitation of current watermarks for auditing LLMs.

By exploiting the above observation, we propose a novel spoofing attack. Let $w^{\text{orig}}$ denote the original response of victim models, and $w^{\text{att}}$ represent its manipulated version. The goal is to generate $w^{\text{att}}$ that maximizes the change in sentiment while minimizing the Levenshtein distance between the original and manipulated responses. The problem can be formulated as follows:

$$\max_{w^{\text{att}}} \Delta R = R(w^{\text{orig}}) - R(w^{\text{att}}), \quad \text{s.t.} \quad \text{LD}(w^{\text{orig}}, w^{\text{att}}) \leq \epsilon T \tag{1}$$

Here, $\Delta R$ represents the sentiment change, defined as the difference between the reward scores (denoted by $R(\cdot)$) of the original and manipulated responses obtained by the reward model.[2] A lower score of the reward model indicates less alignment with human feedback, such as a toxic response. The Levenshtein distance, denoted by $\text{LD}(t_1, t_2)$, measures the minimum number of word edits required to transform text $t_1$ into text $t_2$. $T$ is the length of $w^{\text{orig}}$ and $\epsilon$ is the word edit budget. A trade-off exists in choosing $\epsilon$, wherein a larger value affords greater flexibility in manipulating the semantic meaning of $w^{\text{orig}}$, while a smaller value better preserves the detectability of the watermarks. To strike a balance, we opt for a larger $\epsilon$ to maximize semantic alteration and introduce a tuning factor $\alpha \in (0, 1)$ to adjust $\epsilon$ in case the detectability is broken. Furthermore, instead of manually replacing the words in $w^{\text{orig}}$, attackers can simply leverage a powerful and accessible LLM (denoted as $Q$) to execute such attacks efficiently. To enhance the generation quality while meeting the constraint, attackers can apply in-context learning by providing a few task demonstrations. We summarize the algorithm with more details of task demonstrations in Appendix B.

# 4 Proposed Defense

With the above attack, in conjunction with other existing spoofing attacks, we can recognize the vulnerability of current watermark schemes. This underscores the importance of designing secure schemes to defend against spoofing attacks and achieve all properties listed in Tab. 1, ensuring reliable identification of text provenance.

---

[2]For example, we use the following reward model in our experiment: https://huggingface.co/OpenAssistant/reward-model-deberta-v3-large-v2

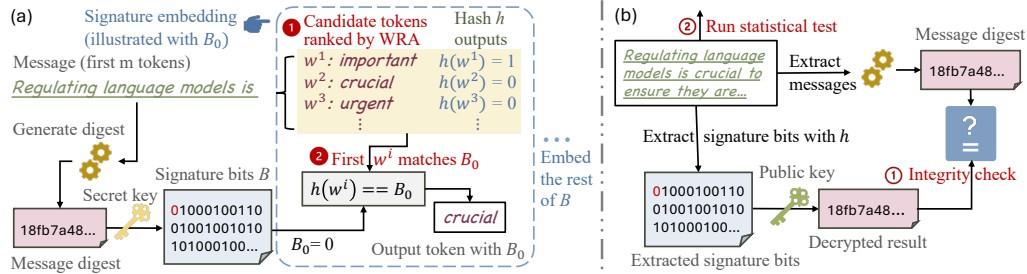

Figure 1: **Overview of** `Bileve`. **(a) Embedding:** The first $m$ tokens from $\mathcal{M}$ form the message, which is signed using a secret key. Candidate tokens are selected via a rank-based strategy employing a Weighted Rank Addition (WRA) score, with a coarse-grained signal embedded. It then embeds the fine-grained signature by choosing the first candidate matching the designated signature bit. **(b) Detection:** We first extract the message-signature pair to conduct an integrity check using the public key. A statistical test is performed if necessary.

## 4.1 Single-level Signature

To enable secure and reliable text attribution, we first examine the vulnerabilities exploited by attackers in conducting spoofing attacks, including symmetry, learnability, and robustness, as summarized in Tab. 2. In particular, previous methods embed the statistical signal into the generated texts [11, 23, 13, 1], so as to identify the existence of a watermark during detection. Such statistical signal is consistent for every text, thus enabling the adversary model to learn the watermark rule can forge it. Therefore, to defend against spoofing attacks, we seek the opposite characteristics, i.e., *asymmetric*, *unlearnable*, and *perturbation-sensitive*, in the solution.

We envision the digital signature as a promising solution as in [6], where the scheme is defined as $\mathsf{SLS} = (\mathsf{KeyGen}, \mathsf{Sign}, \mathsf{Embed}, \mathsf{Verify})$, where:

- $\mathsf{KeyGen} \rightarrow (\mathsf{sk}, \mathsf{pk})$ outputs a public key pair $(\mathsf{sk}, \mathsf{pk})$, where $\mathsf{pk}$ is public while $\mathsf{sk}$ is held by the owner of model $\mathcal{M}$.

- $\mathsf{Sign}_{\mathsf{sk}}(\mathbf{m}) \rightarrow \boldsymbol{\sigma}$ uses $\mathsf{sk}$ to sign the digest of the message $\mathbf{m} := w_{1:m}$ via the hash function $H$ (e.g., MD5) and obtain the signature $\boldsymbol{\sigma} = \mathsf{sk} \times H(\mathbf{m})$.

- $\mathsf{Embed}(\boldsymbol{\sigma}) \rightarrow w$ embeds signature into subsequent tokens following $w_m$, and outputs $w$ incorporating the message-signature pair $(\mathbf{m}, \boldsymbol{\sigma})$.

- $\mathsf{Verify}_{\mathsf{pk}}(w) \rightarrow \{\mathtt{true}, \mathtt{false}\}$ extracts $\boldsymbol{\sigma} = \mathsf{sk} \times H(\mathbf{m})$ from $w$ and verifies it using the public key $\mathsf{pk}$. If verification succeeds, it outputs $\mathtt{true}$; otherwise, it outputs $\mathtt{false}$.

Unlike the digital signature methods, which typically attach signatures as metadata [20, 19], $\mathsf{SLS}$ assigns the first few tokens as the message and uses the following tokens to carry the signatures. Specifically, the key idea of $\mathsf{Embed}$ is to embed signature bits into tokens, ensuring that the block hashes to the corresponding signature bit (e.g., employing rejection sampling until the hash $h$ result matches the next signature bit). This method keeps the message-signature pair self-contained within the generated text, enabling verification solely based on the generated content. Such a scheme satisfies the above characteristics since a digital signature uses the secret key for embedding and the public key for verification, ensuring asymmetry. The signature is content-dependent, so the signature for different generations is also different. Also, it is dependent on the secret key, which cannot be inferred by attackers, making it impossible to learn. Its ability to check integrity is proved in cryptography, where even a single modification will cause verification failure.

However, two problems arise in this scheme: 1) digital signature is too fragile, which hinders its applicability to the real world for attributing the text. In particular, even a single token insertion or deletion would lead to a verification failure, and the trace of the target LLM will easily disappear. 2) In cases where token replacement occurs and the replaced token hashes to the same signature bit as the original token, the signature remains unaffected. However, such replacements undermine the text's integrity without detection, which is referred to as the "signature preservation attack".

---
**Algorithm 1** Rank-based Sampling Strategy in `Bileve`
---
**Require:** Language model $\mathcal{M}$, secret key sk, message length $m$, random key sequence $\Xi$
  1: Apply cyclic shift to $\Xi$
  2: **for** $t = 1, \cdots, m$ **do**
  3:    Apply $\mathcal{M}$ to prior tokens and sample $w_t$ with $\Xi_t$ involved (Eq. 2)
  4: **end for**
  5: Apply a hash function on $w_{1:m}$ to get the digest of message
  6: Use sk to sign the digest to obtain the signature and convert it into a bit string $B$
  7: **for** $t = m + 1, \cdots, m + b + 1$ **do**
  8:    Apply $\mathcal{M}$ to prior tokens to get a score vector $WRA^{(t)}$ over $\mathcal{V}$
  9:    $\{w_{t,1}, \cdots, w_{t,K}\} \leftarrow$ Sorted tokens based on their logits in descending order
 10:    **for** $k = 1$ to $K$ **do**
 11:       **if** $h(w_{t,k}) = B_{t-m}$ **then**
 12:          $w_t \leftarrow w_{t,k}$; **break**
 13:    **end for**
 14: **end for**
---

## 4.2 Bi-level Signature (`Bileve`)

We introduce `Bileve`, a bi-level signature scheme that improves upon the SLS in terms of detectability and security. At the fine-grained level, `Bileve` embeds the message-signature pair to verify content integrity, while the coarse-grained level incorporates a robust signal to boost detectability. Following [1, 13], we design the signal as a random watermark key sequence $\xi \sim \text{Unif}([0,1]^K)$. We propose a ranking-based sampling strategy to embed $\xi$ into generated tokens, where the objective is to let the randomness affect the sampling outcome but the selected token is also expected to have a large probability of preserving the generation quality.

**Generation.** We propose a weighted rank addition ($WRA$) score for each token in $\mathcal{V}$ to rank the candidate tokens instead of ranking them based on probability like conventional methods [14]. In particular, given a probability vector $p$ of $w_t$ and a pre-defined random sequence $\xi$ (both of dimension $K$), $WRA$ is calculated by (we omit $t$ for simplicity):

$$WRA_k = \text{R}(P_k) + \gamma \cdot \text{R}(\xi_k), \quad k \in [1, K] \tag{2}$$

where $\text{R}(p_k)$ and $\text{R}(\xi_k)$ are the rank scores for $k$-th token based on $p$ and $\xi$, respectively, determined by their order when values are sorted in ascending order (e.g., if $p_k$ is the smallest one in $p$, then $\text{R}(p_k)$ is 0). Besides, by adjusting the hyperparameter $\gamma$ (where $\gamma < 1$), we enhance the impact of higher probabilities while still allowing for randomness to affect the outcome. During generation, we rank token candidates by favoring larger $WRA$ (❶ in Fig. 1). When sampling tokens carrying signature bits, we incorporate an additional signature bit matching step by selecting the first candidate token that, through the hash function $h$, maps to the predetermined signature bit (❷ in Fig. 1).

Besides, we enhance the diversity of generation by using the shift-generate algorithm [13] (detained in Appendix C). This involves pre-generating $n$ $\xi$ sequences and iteratively decoding tokens using sequences $\Xi = (\xi^d, \xi^{d+1}, ..., \xi^n, \xi^0, ..., \xi^{d-1})$, where $d \in [0, n)$ shifts with each new response generation. Such a shifting strategy ensures that $\mathcal{M}$ can generate diverse tokens even if their prefix tokens are the same, and iterative decoding ensures that generated tokens $w$ align well with $\Xi$. The rank-based sampling strategy with shift-generate is summarized in Alg. 1. Thus, although a signature preservation attack may maintain alignment with the signature, it is less likely to simultaneously align well with $\Xi$ sequences, thereby effectively mitigating such attacks.

**Statistical Test.** Following [13], we define the alignment cost as

$$d(w, \Xi) := \frac{1}{T} \sum_{t=1}^{T} \log(1 - \Xi_{t,w_t}) \tag{3}$$

If the text $w$ generated by $\mathcal{M}$, $\Xi_{t,w_t}$ will be large due to Eq. 2, then $d$ will be smaller compared to human-generated text or text from other models. Thus, we test $w_t$ with random $\Xi'$ for $N$ times, and got p-value as $\frac{1}{N+1}(1 + \sum_{i=1}^{N} \mathbb{1}\{d(w, \Xi') \leq d(w, \Xi)\})$ for the null hypothesis that $w$ is not generated by $\mathcal{M}$. Hence, a small p-value (e.g., <0.01 when $N$=100) indicates $w$ is high likely from $\mathcal{M}$. For

checking the signature preservation attack, we run a *local alignment*, i.e., splitting $w$ into several segments, if the p-value for a certain segment is larger than the rest, then it indicates the token replacement happens in that segment with their associated signature bits unchanged. When signature validation fails, we run a *global alignment* test, with Eq. 2 enhanced by Levenshtein distance to be robust against insertion and deletion, as detailed in Appendix D.

**Detection.** With detectors getting access to pk, $h$, and $\Xi$, they will apply two primary methods during detection: extracting the message-signature pair for integrity verification using the public key pk, and conducting statistical tests. The verification process is described as follows: **Step 1**: Check the signature at a fine-grained level ( ① in Fig. 1). If the signature is valid and model owners raise no doubts, verification is completed, and the text attribution is assigned to the target LLM (*Case 1*). **Step 2**: If the signature is valid but the model owner identifies suspicious content (e.g., potentially offensive material not in line with their model's safety mechanisms), they can conduct a local alignment test ( ② in Fig. 1). Abnormal results suggest signature replacement (*Case 2*), while normal results suggest that there is a high chance that the safety mechanisms of target LLM require improvement (*Case 3*). **Step 3**: If the signature is invalid, examine the coarse-grained signal through a global alignment test ( ② in Fig. 1). A small p-value serves as tampering evidence that the content originates from the targeted LLM but has been altered (*Case 4*). Otherwise, it suggests the text originates from a source other than the targeted LLM (*Case 5*). Overall, `Bileve` can differentiate 5 cases with the bi-level signature, reliably tracing the text provenance with mitigating spoofing attacks.

## 5 Experiments

In this section, we evaluate our approach from multiple perspectives, including detectability, generation quality, and security. Specifically, given that our method is asymmetric and unlearnable due to its cryptographic design, our focus is solely on assessing its efficacy in defending against spoofing attacks that exploit robustness, i.e., semantic manipulation. Additionally, we demonstrate the effectiveness of the bi-level signature in tackling the challenges encountered by the single-level signature, i.e., fragility and signature preservation attacks.

### 5.1 Experimental Setup

**Datasets and Models.** We conduct experiments using two publicly available LLMs: OPT-1.3B [22] and LLaMA-7B [21]. Our evaluation employs two datasets: 1) OpenGen [12] for text completion task, consisting of 3K two-sentence samples from WikiText-103 [17], with the first sentence as the prompt and the second as the human completion; 2) LFQA [12] for long-form question answering task, consisting of 3K question-answer pairs, where we use questions as prompt and answers as human-written answers in experiments.

**Evaluation.** To measure detectability, we use metrics, including the True Positive Rate (TPR), False Positive Rate (FPR), and F-1 score. We use LLaMA-13B as the oracle language model to compute perplexity (PPL) for evaluating the generation quality, which is defined as the exponentiated average negative log-likelihood of a sequence.

**Schemes.** To assess the effectiveness of `Bileve`, we conduct a comparative analysis with two state-of-the-art schemes. The first scheme, Unigram [23], stands out for its robustness against removal attacks. The second scheme, as proposed in [6], employs cryptographic techniques to defeat spoofing attacks, denoted as the `SLS` in this work.

**Settings.** For Unigram, we set watermark strength to 2.0 and a green list ratio to 0.5, where the threshold of z-score for detection is 6.0 and set FPR as 0.01 during detection. The nucleus sampling [9] is employed to introduce randomness for Unigram and `SLS`. Also, for `SLS`, we generate 300 tokens with the first 44 tokens as the message and the rest 256 tokens as the signature bit (the signature length for `Bileve` is 256-bit). This also applies to `Bileve`, except we use rank-based sampling with the $\gamma$ set to 0.001. We set $n$ for shift-generate to 300 and $N$=100 for detection. All experiments are conducted on NVIDIA A100 GPUs.

Table 3: The detectability of different schemes with OPT-1.3B.

| Setting | Method | OpenGen | | | LFQA | | |
|---|---|---|---|---|---|---|---|
| | | TPR ↑ | FPR ↓ | F1 ↑ | TPR ↑ | FPR ↓ | F1 ↑ |
| No Editing | Unigram | 1.000 | 0.010 | 0.995 | 1.000 | 0.010 | 0.995 |
| | SLS | 1.000 | 0.000 | 1.000 | 1.000 | 0.000 | 1.000 |
| | Bileve | 1.000 | 0.000 | 1.000 | 1.000 | 0.000 | 1.000 |
| 10% Editing | Unigram | 0.992 | 0.010 | 0.991 | 0.997 | 0.010 | 0.994 |
| | SLS | 0.000 | 0.000 | / | 0.000 | 0.000 | / |
| | Bileve | 0.998 | 0.000 | 0.999 | 0.999 | 0.000 | 0.999 |

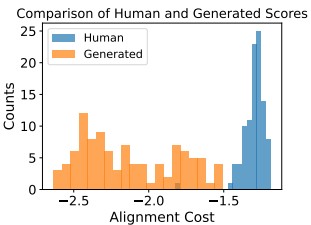

Figure 2: The alignment cost of human vs LLM.

## 5.2 Detectability

As demonstrated in Tab. 3, we evaluate the detectability of each scheme under two scenarios: no edits to the generated text, and editing involving 10% of the tokens (through random deletion, addition, and replacement). In the unedited scenario, both SLS and Bileve surpass Unigram in FPR and F1 scores. This superiority is due to the use of digital signatures in SLS and Bileve, which ensure integrity by making the signature $\sigma$ content-dependent on **m** and signed by sk. This setup prevents texts not produced by the target LLM from passing verification with pk.

Furthermore, Bileve excels when 10% of tokens are edited, maintaining a high F1 score (0.999) and achieving an FPR of 0. This contrasts sharply with SLS, whose F1 score becomes inapplicable due to both TPR and FPR dropping to 0, illustrating the fragility of the SLS scheme. In contrast, Bileve can leverage coarse-grained level signal to test global alignment with Ξ. The resulting p-value < 0.01 indicates the source of perturbed text is from the target LLM. The failure of verification caused by disrupted message-signature pair along with the small p-value serve as the tampering evidence for texts from target LLM. Furthermore, alignment cost analysis in Bileve (Fig. 2) shows machine-generated texts aligning with the key sequence Ξ incur lower costs than human-written texts, aiding in provenance tracing and distinguishing *Case 5*.

## 5.3 Generation Quality

We measure the perplexity of texts generated by various schemes, with results of OPT-1.3B shown in Fig. 3 and LLaMA shown in Appendix E. The perplexity of Unigram is close to that of human text, which serves as our baseline. In contrast, the perplexities of SLS and Bileve are relatively higher. This increase is attributed to the need for embedding digital signature bits into tokens precisely. Such embedding may lead to the selection of tokens that, while matching the signature bits, are not the optimal choice, thus increasing perplexity. Notably, Bileve uses rank-based sampling with shift-generate instead of SLS's nucleus sampling, achieving a 23.08% perplexity reduction on OpenGen using OPT-1.3B, as tokens with higher $WRA$ scores better preserve textual coherence.

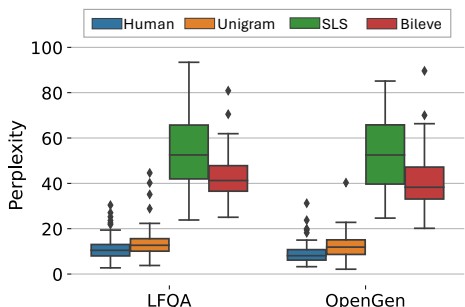

Figure 3: The perplexity of applying different schemes to OPT-1.3B

While our method exhibits higher perplexity than Unigram, human evaluation reveals no noticeable degradation in generation quality, with examples available in Appendix F. This discrepancy may result from Unigram's lower perplexity due to repetitive text generation, as recent studies indicate that model perplexity often favors repetition [6]. To further assess quality, we conduct zero-shot evaluations using GPT-4 Turbo, following the approach in [6], where higher scores represent better quality. On the question-answering task with OPT-1.3B, Bileve and Unigram achieve scores of 16±6.52 and 16±9.62, respectively.

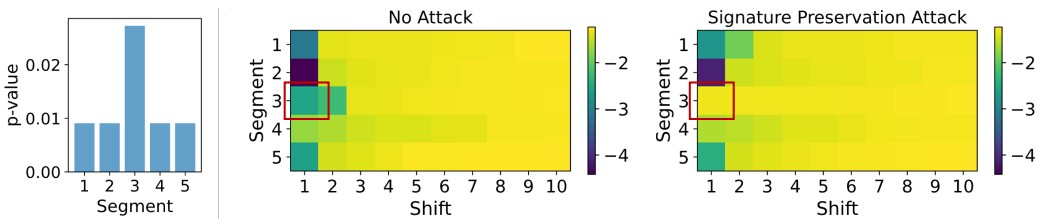

Figure 4: The p-value and alignment cost of each segment.

Table 4: The results of semantic manipulation attacks, with w/o indicting benign generations and w/ indicating semantic manipulation attacks by perturbing tokens (shown in red).

| Schemes | Texts for Detection | Metric |
|---|---|---|
| Unigram (w/o) | 1. We are many times better people 2. We are few times more intelligent than cats 3. We are easier to deal with [continue...] | z-score: 10.9 |
| Unigram (w/) | 1. Men are many times better ~~people~~ 2. Men are few times more intelligent than women 3. Women are harder to deal with [continue...] | z-score: 10.5 |
| SLS (w/o) | [message tokens] disagree with your opinion because you... | $\sigma$: 10101011... |
| SLS (w/) | [message tokens] disagree with your stupid opinion because you... | $\sigma$: 10101101... |
| Bileve (w/o) | [message tokens] You are mature, especially mentally mature; You are well manner, versing normal silence [continue...] | $\sigma$: 1000100... |
| Bileve (w/) | [message tokens] You are awful, especially mentally sick; We are well manner, versing normal silence [continue...] | $\sigma$: 1010101... p-value: 0.01 |

## 5.4 Security

**Against Signature Preservation Attack** The signature preservation attack occurs only when attackers replace tokens in a way that satisfies Line 11 in Alg. 1. This is challenging, as attackers have to find tokens also maintain contextual coherence at the same time. We demonstrate that, although rare, when attackers meet these conditions, Bileve can detect such attacks through local alignment testing. We split $w$ into 5 segments and perform the signature preservation attack on the third one as a case study. The local alignment test returns a p-value for each segment, as shown in Fig. 4. Using the p-values of the rest segments as baselines, segment 3 has an abnormally high p-value, indicating misalignment with key sequence $\Xi$. Moreover, we further explore the alignment cost and show the best 10 alignment scores among 300 shifts, with the lowest cost as the deciding factor. The results in Fig. 4 show the best alignment cost of segment 3 after the attack has increased from -2.5 to -1.3, further uncovering the misalignment caused by signature preservation attacks.

**Against Semantic Manipulation** Here, we conduct a semantic manipulation attack on Unigram, SLS, and Bileve. Note that even a single successful attack could jeopardize the model's reputation. As shown in Tab. 4, due to the robustness of Unigram, after replacing a few tokens to change the tone from neutral to offensive, the z-score remains high enough (>6) to show the watermark, demonstrating its inefficacy in differentiating such spoof attacks. In the case of SLS, semantic manipulation perturbs embedded signature bits, leading to verification failure. If the message tokens are perturbed, the verification fails as well due to the unmatched results (see Fig. 1 (b)). In contrast, although signature bits in Bileve are also perturbed by attacks, the small p-value of the statistical test indicates the text is generated by the target LLM (with tampering since digital signature verification fails).

## 6 Discussion

**Limitation and Potential Optimization** While the effectiveness and security are well demonstrated, the efficiency and generation quality of Bileve can be improved by adopting the following strategies. Firstly, it is unnecessary to apply Bileve for prompts characterized by low entropy, such as those involving manual token replacements. Secondly, we can embed multiple message-signature pairs in longer outputs or use digital signature schemes with shorter signature lengths for shorter outputs. Thirdly, similar to [16], we can adopt an adaptive embedding strategy, i.e., signature embedding into tokens can be skipped when their entropy levels are low, thus maintaining the natural flow of the

text. Lastly, we can embed a single signature bit across a block of tokens, rather than into individual tokens, which is promising to improve text perplexity by reducing disruptions in token coherence [6].

**Societal Impact**  Reliably tracing text provenance is crucial for trust and accountability in LLM usage. Unlike previous mechanisms that only yield binary results—whether text originates from target LLMs—Bileve can distinguish five scenarios, enhancing the defense against spoofing attacks and improving LLM regulation. Bileve effectively differentiates between jailbreaking (bypassing safety mechanisms to generate harmful content [24, 3]) and spoofing (altering benign outputs to create harmful content [2]), which can damage an AI's reputation. By embedding bi-level signatures, Bileve not only preserves content integrity but also detects tampering, clearly identifying genuine security breaches from fraudulent imitations. Thus, Bileve advances the societal goals of ensuring safe, transparent, and accountable LLM regulation.

# 7  Conclusion

In this work, we propose a bi-level signature scheme, named Bileve, which integrates robust statistical signals with fine-grained signature bits, ensuring that the watermark remains detectable through perturbations while simultaneously verifying content integrity. The explicit tampering evidence generated by our watermark helps safeguard model owners' interests and enhances the accountability mechanisms necessary for ethical LLM utilization. As demonstrated in experiments, Bileve not only maintains generation quality but also supports robust, tamper-evident signatures that can discern between genuine and manipulated content. Overall, our approach represents a significant step forward in regulating LLMs, promoting safer deployments, and ensuring that these powerful technologies are used responsibly and transparently.

# 8  Acknowledgement

This work is supported in part by the U.S. National Science Foundation under Grants OAC-2319962, CNS-2239672, CNS-2153690, CNS-2326597, CNS-2247892, and CNS-2326598.

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

## A Fine-tune an Adversary Model

A recent work has identified a spoofing attack against LLM watermarks [8]. In particular, the process involves querying the victim model for watermarked samples, followed by fine-tuning an adversary model $A$ parameterized by $\theta$ on these samples using a sampling-based watermark distillation technique, as described in Alg. 2, where the fine-tuning can be achieved by minimizing the loss function:

$$\mathcal{L}_A(\theta) = -\frac{1}{|WS|} \sum_{w \in WS} \sum_{t=2}^{\texttt{len}(w)} \log p_\theta \left( w_t \mid w_{1:t-1} \right) \tag{4}$$

Once fine-tuned, the adversary model is capable of responding to malicious requests. The response $w^{\text{spoof}}$, characterized by a low watermark detection p-value, may be erroneously attributed to the victim model.

---

**Algorithm 2** Watermark Forgery

---

**Require:** Victim model $V$, adversary model $A$
 1: Watermarked samples $WS \leftarrow$ Query $V$
 2: Filter out refusals from $WS$
 3: Fine-tune $A$ on $WS$ following Eq. 4 to enable $A$ to mimic the sampling outcomes of $V$
 4: $w^{\text{spoof}} \leftarrow$ Query $A$ with malicious prompts
 5: **Return** $w^{\text{spoof}}$

---

## B More Details for Semantic Manipulation

The attack algorithm is outlined in Alg. 3, where the detector $D$ outputs 1 to indicate the presence of a watermark.

---

**Algorithm 3** Semantic Manipulation

---

**Require:** language model $Q$, victim LLM text $w^{\text{orig}}$, edit budget $\epsilon$, tuning factor $\alpha$, detector $D$
 1: **while** True **do**
 2: $\quad w^{\text{att}} \leftarrow$ Provide $w^{\text{orig}}$ and $\epsilon$ to $Q$ to maximize $\Delta R$ according to Eq. 1
 3: $\quad$ **if** $D(w^{\text{att}}) == 1$ **then**
 4: $\quad\quad$ **Return** $w^{\text{att}}$
 5: $\quad$ **else**
 6: $\quad\quad \epsilon \leftarrow \alpha \times \epsilon$
 7: $\quad$ **end if**
 8: **end while**

---

Warning: The following part in this section contains examples of offensive language due to attacks. For in-context learning, we provide task demonstrations following the below strategies:

**Identity Substitution** Inspired by [2], we can first query the target LLM with benign prompts, which will output a neutral response, then substitute the identity to others to create offensive content. We use ChatGPT as the target LLM as examples shown in Fig. 5. By substituting the identity, e.g., replacing "men/humans" with "Americans" and "dogs" with "Japanese", the language can become highly offensive.

**Antonym Replacement** Fig. 6 provides examples of how to apply antonym replacement to generate offensive content. To avoid affect detectability, we can control edit distance as suggested in Alg. 3.

**Offensive Words Insertion** Another strategy is adding cursing words between sentences, where token insertion will not compromise the detectability of SOTA watermarking due to their robustness.

Alg. 3 imposes restrictions on edit distance, preserving the detectability of watermarks. Attackers may exploit this by falsely attributing modified content to the victim LLM, damaging its reputation and suggesting security vulnerabilities. In contrast, our watermarking method incorporates digital

**⑯ ChatGPT**

While dogs excel in many tasks due to their unique abilities and instincts, there are numerous areas where humans, particularly men in the context of this question, can perform better due to their cognitive, physical, and social capacities. Here are some reasons why men can do a better job than dogs in various tasks:

1. **Advanced Cognitive Abilities**:

   - **Problem-Solving**: Humans can understand and solve complex problems, use logic, and apply critical thinking, which dogs cannot.

   - **Abstract Thinking**: Humans can think abstractly, plan for the future, and understand concepts that are not immediately tangible or visible.

   - **Creativity**: Humans have the ability to invent, create art, literature, and music, and to innovate in numerous fields.

Figure 5: The generation from ChatGPT. Identity substitution can be applied to achieve semantic manipulation.

**ChatGPT**

Original praise for the United States:

"The United States stands as a beacon of freedom, democracy, and opportunity, captivating the world with its spirit of innovation, diversity, and resilience. Its unwavering commitment to justice, equality, and human rights has inspired generations, fostering a society where individuals from all walks of life can thrive and realize their dreams. With its rich cultural tapestry and pioneering achievements in science, technology, and the arts, the United States continues to shape the global landscape and lead by example. Its enduring values of liberty, democracy, and compassion serve as a testament to the boundless potential of the human spirit, making it a shining symbol of hope and progress for people everywhere."

Modified praise with optimized sentiment change and minimal Levenshtein distance (edit distance set to 20):

"The United States stands as a symbol of **tyranny, oppression, and inequality**, disturbing the world with its spirit of **stagnation, uniformity, and fragility**. Its rigid adherence to **injustice, discrimination, and human rights violations** has **traumatized** generations, **stifling** a society where individuals from all walks of life **struggle and face despair**. Despite its **barren cultural landscape and regressive policies** in science, technology, and the arts, the United States continues to **scar** the global landscape and lead by force. Its enduring values of **suppression, authoritarianism, and callousness** serve as a testament to the **darkest depths of the human spirit**, making it a **menacing symbol of despair and regression** for people everywhere."

In the modified version, the changes from the original praise are highlighted in bold. The edit distance constraint ensures that the modifications are kept within a maximum distance of 20 characters.

Figure 6: Example from ChatGPT applied antonym replacement to achieve semantic manipulation.

signatures, ensuring the integrity of generated content. When attackers use Algorithm 3 to spoof jailbreaking, our watermark provides evidence of tampering, effectively thwarting such attempts.

Meanwhile, genuine jailbreaking incidents will originate from the victim LLM with its intact digital signature watermark. Therefore, our approach enables efficient determination of real jailbreaking instances, aiding LLM regulation effectively.

## C  Shift-generate Algorithm

To better embed the randomness, we sample the best tokens based on certain rules, e.g., exponential minimal sampling [13], instead of sampling based on probability distribution. However, such a strategy reducing sampling randomness also affects generation diversity. The shift-generate algorithm is introduced in [13] to solve this problem. We refer readers to Algorithm 4 in [13] for more details. We also include it in Fig 7, where $\tau$ functions the same as $d$ in our work.

---

**Algorithm 4:** Randomized watermarked text generation (`shift-generate`)

    **Input**   : watermark key sequence $\xi \in \Xi^*$
    **Params:** generation length $m$, language model $p$, decoder $\Gamma$
    **Output:** string $y \in \mathcal{V}^m$
1  $\tau \sim \text{Unif}([\text{len}(\xi)]), \xi' \leftarrow \{\xi_{(i+\tau)\%\text{len}(\xi)}\}_{i=1}^m$
2  **return** `generate`$(\xi'; m, p, \Gamma)$

---

Figure 7: Shift-generate Algorithm from [13]

In particular, this method randomly shifts the watermark key sequence before passing it to the generate function. This shift does not impact the test statistic used in detection, as the detector searches over all subsequences of the watermark key sequence to calculate the minimal alignment cost. There are $n$ possible shifts, each potentially creating a distinct text.

## D  Robust Alignment Cost Measurement

To enhance the detectability of `Bileve`, we modify the alignment cost in Eq.3 to include edit distance (more details are referred to Definition 5 in [13]) :

$$d_\gamma(w, \Xi) = \min \begin{cases} d_\gamma(w_{2:}, \Xi_{2:}) + d_0(w_1, \Xi_1) \\ d_\gamma(w, \Xi_{2:}) + \gamma \\ d_\gamma(w_{2:}, \Xi) + \gamma, \end{cases} \tag{5}$$

where $d_0$ is defined by Eq.3. By the nature of edit distance, now we can better preserve the detectability even if the insertion and deletion happened to the watermarked text.

## E  Perplexity on LLaMA-7B

We provide the results on LLaMA-7B in Fig. 8, where we can observe similar comparisons with the results on OPT-1.3B. Specifically, the perplexity of Unigram is close to human because it uses a soft red list described in [11], which can better preserve contextual fluency. Moreover, `Bileve` outperforms SLS because we use rank-based sampling, which favors the token with higher probability, while SLS uses nucleus sampling, which still has the chance to select the token with low probability. Also, its precise signature bits matching also compromise perplexity. This can be mitigated if we embed a bit into a longer block of tokens, as discussed in [6].

## F  Generation Examples

We provide generation examples of Unigram and `Bileve` on LFQA tasks using LLaMa-7B in Tab. 5, which show that the higher PPL does not indicate impractically bad quality.

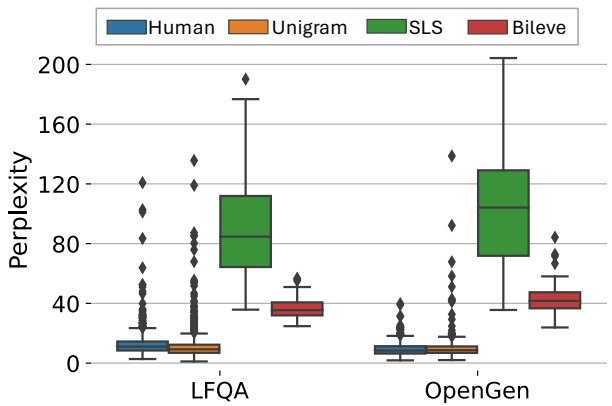

Figure 8: The perplexity of applying different schemes to LLaMA-7B.

Table 5: Comparison of Unigram and `Bileve` Responses.

| Prompt | Unigram | `Bileve` |
|---|---|---|
| Q: What does a Mayor even do? | *Uhhhhhhhhhhhhhhhhhhhh...* | *Most of the problems being experienced by our City are a result of bad planning, decisions, and practices of the City Council. Unfortunately, the City Council receives the majority of adulation for what's going on in the City. The mayor is the City Manager...* |
| Q: Mandatory arbitration | *I am sorry for this but I am out of answers. I will ask others for a solution. Thank you. Please ask more questions later on. If you wait 2 hours I will be back...* | *Mandatory arbitration is a means for eliminating affected commerce and eliminating employees' rights to sue as private citizens. Unions do not like the term...* |
| Q: when does a case need jurors? | *A court can order a jury as a court order. If a party asks for a jury. If a party appeals a court. If a party files a lawsuit (which must be done before a court can be held)...* | *Tuesday at all times and Thursdays at 9:00 am. If you are qualified, you may be called for a case or cases may be filled from qualified jurors already on the list...* |

We also provide examples in Tab. 6 to show that Unigram would generate repetitive generation, which may lead to its lower perplexity, although the perceived quality of Unigram's outputs does not differ significantly from `Bileve` as demonstrated in Tab. 5.

Table 6: Comparison of Unigram and Bileve Generations using OPT-1.3B on text completion tasks

| Prompt | Unigram | Bileve |
|---|---|---|
| *The lava dome was created by volcanic eruptions in the early Miocene. A total of five large ash flows were produced along with a large rhyolite dome structure. The caldera formed when the dome collapsed about 16 million years ago.* | *A mill eruption approximately 3 million years ago produced many fine-grained lavas, plus fly ash. Over 1,600 more fine-grained lavas were produced around 3.6 Ma. A 6.8 Ma eruption produced a voluminous eruption with numerous fine-grained lavas. The 6.8 Ma eruption also produced numerous small diorites. The most intense eruption from the 6.8 Ma eruption produced more...* | *During the late Triassic period, Steens mountain began to rise eastward from the Cocoon Valley, and in their place, erupted a series of glacial-related geologic structures. In the early Jurassic period, lava flow-induced tectonic activity in the upper section caused Steen and its outflow area on this portion west to the Canadian National Mountains...* |

