# OpenReview forum: "Bileve: Securing Text Provenance in Large Language Models Against Spoofing with Bi-level Signature"
_NeurIPS.cc/2024/Conference — NeurIPS 2024 poster_

### Official Review · Reviewer_6UZB · 2024-07-09

**Soundness:** 3
**Presentation:** 3
**Contribution:** 3
**Rating:** 6
**Confidence:** 3

**Summary:**

The robustness of previous watermark algorithms would lead to a type of spoofing attack where attacker would modify the watermarked text to contain harmful contents while ensuring the watermark can still be detected. This paper introduce a bi-level signature scheme called bileve to mitigate spoofing attacks and enhance detectability. Bileve could recognize 5 scenarios during detection, compared to only 2 for previous methods. Using bileve, the LLM owners could verify if the source of the given texts. From experiments, the effectiveness of bileve against spoofing attack is validated.

**Strengths:**

1. The 3 types of spoofing attacks are clearly listed in this paper, with exploited vulnerabilities explained, which makes the motivation reasonable.
2. The paper provides comparsion between single-level signature and bileve, which is good for understanding.
3. Bileve could produce a total of 5 different detection results, which meets real-world cases.

**Weaknesses:**

1. Although with the proposed WRA, the text quality is improved as compared to SLS, the difference between bileve and unigram is still noticable.
2. For case 4&5, if the watermarked text is inserted into a long document (copy-paste attack), then the global alignment test would not produce a small p-value while the detected text does contain watermarked text.

**Questions:**

1. what is the generation/detection complexity? The statistical test during detection seems quite time-consuming.
2. Are all 5 cases tested during evaluation? Details of how local alignment test is conducted, e.g., chunk size, can be clarified in the paper.

**Limitations:**

The paper could provide more info on the complexity of the generation and detection process.

---

> ### Author Rebuttal · Authors · 2024-08-04
>
> Thank you for your questions and feedback. We appreciate the opportunity to address your concerns here.
>
> **W1: Quality**
>
> The key baseline of this work is SLS, as it also focuses on defeating spoofing attacks. Bileve has already shown improvements over SLS, indicating potential for further enhancements. This highlights the feasibility of balancing security and quality in future developments. It is important to note that Bileve is primarily focused on establishing a secure foundation. We plan to enhance text quality while maintaining these security features in future work, with potential optimizations discussed in Section 6.
>
> Additionally, the lower perplexity of Unigram may result from repetitive text, where model perplexity favors repetitive text as discussed in recent research  [1].
> For example, for text completion tasks with prompt
> >The lava dome was created by volcanic eruptions in the early Miocene. A total of five large ash flows were produced along with a large rhyolite dome structure. The caldera formed when the dome collapsed about 16 million years ago.
>
> Generation of Unigram using OPT-1.3B:
> >A mill eruption approximately 3 million years ago produced many fine-grained lavas, plus fly ash. Over 1,600 more fine-grained lavas were produced around 3.6 Ma. A 6.8 Ma eruption produced a voluminous eruption with numerous fine-grained lavas. The 6.8 Ma eruption also produced numerous small diorites. The most intense eruption from the 6.8 Ma eruption produced more fine-grained lavas, with many made of pyroxene, basalt, and calcrete...
>
> Generation of Bileve using OPT-1.3B:
> >During the late Triassic period, Steens mountain began to rise eastward from the Cocoon Valley, and in their place, erupted a series of glacial-related geologic structures. In the early Jurassic period, lava flow-induced tectonic activity in the upper section caused Steen and its outflow area on this portion west to the Canadian National Mountains...
>
> Thus, we plan to incorporate other evaluation metrics, such as human evaluation, in the revised paper to provide a more comprehensive assessment of text quality.
>
> **W2: Potential Attack**
>
> Thanks for pointing out this potential attack. Our work aims to defeat spoofing attacks. In the copy-paste attack you mentioned, it cannot achieve spoofing attacks, i.e., faking the long document as being generated by the target LLM. If a long document with only a small portion of watermarked content is fed into our detector, our detector would not determine that this long document was generated by the target LLM, which makes sense.
>
> Our scheme currently does not differentiate between fully watermarked documents and those with only partial watermarked content. However, in cases where this differentiation is desired, we can perform a linear search along the document. For documents longer than the key sequence, we can check if there is a segment of the document that aligns well with the key sequence. For documents shorter than the key sequence, we can run a local alignment. Unlike case 2 and the results in Figure 4, this approach would result in the p-value being small for the watermarked segment while being large for other parts of the document.
>
>
> **Q1: Generation/Detection Complexity**
>
> Generation complexity is linear in the length of the input text. During token generation, we need to sample a token that matches the signature bit, which can increase the runtime by up to three times compared to generating tokens without matching signature bits.
>
> The statistical test during detection follows the design in [1], and its computational complexity is linear in the length of the watermark key sequence. However, these statistical tests are only employed when the signature is invalid (i.e., in the event of an attack). Under normal circumstances, we simply verify the validity of the signature.
>
> **Q2: Evaluation**
>
> The results of case 1 and case 4 can be found in Table 3, and case 2 in Figure 4. Case 3 involves unexpected content generated by the target LLM, potentially due to jailbreaking, and is discussed in the discussion section and Appendix B. Case 5 indicates content generated by another source, as it does not use the secret key or key sequence during generation.
>
> For the local alignment test, the number of segments is a hyperparameter set to 5 in our experiments. This setting involves a trade-off: a larger number of segments may overlook local realignment, while a smaller number increases latency overhead. We will add a detailed discussion of this trade-off in the revised version of the paper.
>
> ---
> [1] Publicly Detectable Watermarking for Language Models, arxiv 2024.
>
> [2] Robust Distortion-free Watermarks for Language Models, TMLR 2024.

---

> > ### Comment · Reviewer_6UZB · 2024-08-08
> > **reply**
> >
> > Thank you for your reply and explanation, I will maintain my rating.

---

> > > ### Author Response · Authors · 2024-08-10
> > >
> > > Thank you for acknowledging our rebuttal. If you have any further concerns, please let us know. We would like to address them before the discussion period ends.

---

### Official Review · Reviewer_ezrE · 2024-07-11

**Soundness:** 3
**Presentation:** 3
**Contribution:** 3
**Rating:** 7
**Confidence:** 4

**Summary:**

The paper presents a novel approach to secure the provenance of texts generated by large language models (LLMs) through a bi-level signature scheme. This method aims to mitigate spoofing attacks—where malicious actors alter the content generated by LLMs to forge harmful content or misattribute blame—by integrating fine-grained signature bits for integrity checks and a coarse-grained signal for source tracing when signatures are invalidated.

**Strengths:**

1. This paper reveals a spoofing attack that takes advantage of the robustness features of state-of-the-art watermarking schemes.

2. This paper improves the ability to trace the provenance of text and regulate LLMs by differentiating between five detection scenarios.

3. This paper introduces a novel bi-level signature scheme that enhances text provenance security for large language models (LLMs). It combines fine-grained signatures for integrity checks with coarse-grained signals for source tracing.

**Weaknesses:**

1.  While the experiments demonstrate effectiveness in specific settings with OPT-1.3B and LLaMA-7B models, the generalizability and scalability of the Bileve scheme to other models are somewhat uncertain. Authors could consider using larger or more powerful LLMs to demonstrate the effectiveness of the proposed algorithm.

2. The authors could consider using a more powerful LLM to measure the perplexity, like GPT-3/GPT-4.

3. I suggest reporting TPR scores at fixed low FPR (FPR = 10% or 1%).

4. This paper demonstrates detectability by modifying 10% of the tokens. It would be good to test with a higher rate of token modification, like 20%, 30%, to further validate the detectability.

**Questions:**

Please see above.

**Limitations:**

Please see above.

---

> ### Author Rebuttal · Authors · 2024-08-03
>
> Thanks for your feedback, and we would like to address the weakness (**W**) below.
>
> **W1: Generalizability**
>
> The primary goal of our experiments was to establish a proof of concept for the Bileve scheme. We believe demonstrating effectiveness with models like OPT-1.3B and LLaMA-7B would provide a solid foundation for further research. **It is worth noting that Bileve is designed to be generalizable to any auto-regressive LLMs, as the core mechanism is not inherently tied to specific models or affected by model size.** Therefore, we anticipate similar results regarding security with larger models. However, we plan to explore the generation quality when this scheme is applied to more powerful LLMs in future work.
>
> **W2: Perplexity Measured by More Powerful LLMs**
>
> We used LLaMA-13B to measure perplexity since it is an open-sourced powerful LLM, and it outperforms GPT-3 on most benchmarks [1]. Moreover, it is worth noting that **close-sourced models like GPT-4 cannot be used to evaluate the perplexity of watermarked text since it does not return the probability of watermarked tokens.**
> However, we would like to introduce other evaluation metrics that can more comprehensively evaluate the generation quality, like zero-shot quality measurements with GPT-4 Turbo, as suggested in recent research [2]. This metric uses GPT-4 to evaluate the coherence and relevance of generated text, and can be used as a supplemental metric to perplexity.
>
> **W3: Fixed FPR**
>
> In our work, we provided a solution to "how to avoid an LLM being wrongly blamed" by adapting digital signatures, a cryptographic mechanism that ensures an FPR of 0 (as reported in Table 3). As shown in L174 and Figure 1(b), verification only outputs True if the decrypted result using the public key matches the message digest. This verification process relies on computational matching, and digital signatures make it computationally infeasible for unauthorized entities to produce a valid message-signature pair without the private key, thus eliminating false positives.
>
> As a result, we cannot set or report FPR values of 10% or 1%, as our scheme's design inherently prevents false positives. This differentiates it from other methods [3,4] where FPR can be adjusted or measured independently.
>
> **W4: Higher Rate of Token Modification**
>
> We provide more results with higher rates of token modification below on OPT-1.3B with two tasks, and we report the performance using the true positive rate (TPR), where a higher TPR indicates better robustness against removal attacks.
>
>
> When the editing ratio is 20%:
> |          | OpenGen  | LFQA    |
> |----------|----------|---------|
> | Unigram  | 0.979   | 0.971  |
> | Bileve   | 0.989   | 0.987  |
>
> When the editing ratio is 30%:
> |          | OpenGen  | LFQA    |
> |----------|----------|---------|
> | Unigram  | 0.958   | 0.925  |
> | Bileve   | 0.960   | 0.970  |
>
> From the above tables, we can see that although the FPR decreases as the editing ratio increases, Bileve still outperforms Unigram, which can be attributed to the introduced alignment with key sequences. We will conduct a more thorough evaluation with varying edit ratios in the revision.
>
> ---
> [1] LLaMA: Open and Efficient Foundation Language Models, arXiv, 2023
>
> [2] Publicly Detectable Watermarking for Language Models, arxiv 2024.
>
> [3] A Watermark for Large Language Models, ICML 2023.
>
> [4] Provable Robust Watermarking for AI-Generated Text, ICLR 2024.

---

> > ### Comment · Reviewer_ezrE · 2024-08-10
> >
> > Thank you for your clarification. I have raised the score to 7. If possible, please incorporate the extra experimental results into the next version of the paper.

---

> > > ### Author Response · Authors · 2024-08-10
> > >
> > > Thank you for your thorough review and valuable feedback on our work. We will incorporate additional experimental results in the revised version.

---

### Official Review · Reviewer_4vVx · 2024-07-12

**Soundness:** 1
**Presentation:** 1
**Contribution:** 2
**Rating:** 3
**Confidence:** 4

**Summary:**

This paper proposes to consider spoofing attack, where an attacker wants to prove the proposition like "The person holding this watermark private key used an LLM to write this text A." where text A is constructed by the attacker. The paper proposes a defense against spoofing attacks.

**Strengths:**

This paper points out the fundamental trade-off between defending against removal attacks and spoofing attacks.

**Weaknesses:**

I have doubt on the significance of spoofing attack. It is important to first clarify a potential misunderstanding. Authors may believe that a watermark in the text proves "the person holding this watermark private key used an LLM to write this text A." But that's not accurate.

However, the watermark only proves that the probability of text A being generated by a process independent of the watermark key holder is very low. It does not conclusively prove the key holder generated that specific text A.

Therefore, I believe the spoofing attack lacks real significance from the outset. If someone wants to prove they said certain things and nothing else, they can just use a traditional digital signature.

The problem is also framed as "How to avoid an LLM being wrongly blamed?" But what can we really blame an LLM for? Sure, there may be instances where a single LLM inference generates a token sequence that is interpreted as harmful by humans.

However, LLMs are probabilistic models that can potentially generate any harmful content given enough inferences. We can only blame an LLM for having a high average probability of generating harmful content, not for the existence of individual harmful inferences.

Moreover the paper appears hard to read to me. For example, "instead of ranking them based on probability like conventional methods [13]" doesn't specify what conventional methods mean in paper [13] Pre-trained language models for text generation: A survey.

Furthermore, t's unclear if the signature preservation attack requires constructing two messages with the same hash, as implied by "replaced token hashes to the same signature bit." If so, that would be extremely difficult for modern hash functions.

More importantly, the paper does not provide any rigorous theoretical guarantees that Bileve actually solves the spoofing attack issue as claimed. The key assertion is that "it is less likely to simultaneously align well with $\Xi$ sequences, thereby effectively mitigating such attacks." However, this statement is quite vague and unconvincing on its own.

What does it mean for a method to be "less likely to simultaneously align well with $\Xi$ sequences"? How much less likely is it quantitatively? Under what assumptions or conditions does this property hold? The paper does not provide clear answers to these crucial questions.

**Questions:**

In the definition of "signature preservation attack", I saw that "replaced token hashes to the same signature bit". Does it mean that signature preservation attack requires constructing two messages with the same hash? If so, that would be extremely difficult for modern hash functions.

**Limitations:**

This paper is difficult to read, e.g. the reference to "conventional methods" in "[13]" is unclear.

Does not provide clear theoretical guarantees that the Bileve method effectively mitigates spoofing attacks as claimed.

---

> ### Author Rebuttal · Authors · 2024-08-03
>
> Thanks for your comments and we would like to address the weaknesses (**W**) and questions (**Q**) individually.
>
>
> **W1**
>
> >I have doubt ... specific text A.
>
> First, we understand that you are emphasizing the possibility of false positives in watermark detection, suggesting that non-watermarked text could be incorrectly identified as watermarked. Thus, you question the necessity of conducting spoofing attacks and our motivation to counter them. However, researchers in this field have acknowledged the importance of maintaining a low false positive rate (FPR), which is why learning-based detectors are criticized for their high positive rates and why OpenAI shut down their official detector​ [1]. **In contrast, watermarking has been envisioned as a more reliable and promising method to identify the source of LLM-generated text [2,3]. When no attacks happen, false positive detections are statistically improbable [2].** As a result, people are prone to believe that the text is from a specific LLM (associated with a watermark key) if the watermark is detected. Leading companies like Google have already deployed watermarks in their products to identify AI-generated content [4].
>
> **However, spoofing attacks have unveiled the vulnerability that existing watermarks are not as reliable as previously thought.** In such attacks, attackers can create content that makes people believe it is from the target victim LLM. When done on a large scale, this invalidates the value of the watermark and may cause reputational damage to the model owner (e.g., if inappropriate text is incorrectly attributed to them). **Spoofing attacks have been identified in several works (listed in Table 2), and their significance is acknowledged by many [3], including the rest of the reviewers.**
>
>
> **W2**
> >If someone wants to prove they said certain things and nothing else, they can just use a traditional digital signature.
>
> We would like to clarify that **the scope of this work is focused on reliable LLM detection, rather than verifying if someone said certain things.** Our method adapts digital signatures into LLM generation to achieve reliable detection. We would appreciate more details on how a traditional digital signature can be used to achieve this goal within the context of LLM-generated content, if not through our approach.
>
> **W3**
> >The problem is also framed ... harmful inferences.
>
> When we talk about blaming an LLM, we are essentially addressing the responsibility of the model owner in preventing the LLM from producing abusive or harmful content. If individual harmful inferences were not a concern, there would be no need for red-teaming [5] to ensure the safety of LLMs. These practices are employed precisely because preventing the generation of harmful content is crucial for the responsible deployment of LLMs.
>
> **W4**
> >More importantly, ...crucial questions.
>
> **It is important to note that Bileve adopts digital signatures to combat spoofing attacks.** Specifically, it is computationally infeasible for an attacker who only knows the public key to infer the secret key or produce a valid message-signature pair. For example, the security of RSA digital signatures is based on the difficulty of factoring large composite numbers, with the factorization of a 1024-bit RSA key requiring several years with distributed computing resources. The theoretical foundation of digital signatures is well-established, as discussed in Section 3.2 of [6].
>
> The phrase "less likely to simultaneously align well with sequences" is associated with the case of "signature preservation attacks," which are adaptive attacks against Bileve. We use the term "less likely" because, while our experiments (see Section 5.4) demonstrate that our method can effectively defeat these adaptive attacks, this is empirical evidence. Also, we repeat this attack for 100 times and none of them simultaneously align well with sequences. Thus, this term is used to reflect the current state of our findings and the inherent challenges in providing absolute theoretical guarantees against all potential future attacks. We would like to rephrase it as "it is resistant to simultaneously aligning well with $\Xi$ sequences".
>
> **Q: Signature Preservation Attack**
>
> Here we do not mean that attackers would construct two messages with the same hash. In our scheme, the watermarked content consists of a message-signature pair. This attack considers the scenario where attackers only modify the signature part. Since signature bits are associated with tokens (mapped by function h, as shown in Fig 1), it is possible that the replaced token results in the same signature bit as before. This is why we refer to it as a signature preservation attack.
>
> ---
> [1] https://openai.com/index/new-ai-classifier-for-indicating-ai-written-text/
>
> [2] A Watermark for Large Language Models, ICML 2023.
>
> [3] Watermark Stealing in Large Language Models, ICML 2024.
>
> [4] https://deepmind.google/discover/blog/watermarking-ai-generated-text-and-video-with-synthid/.
>
> [5] Red Teaming Language Models with Language Models, EMNLP 2022.
>
> [6] An Efficient Signature Scheme from Bilinear Pairings and Its Applications, PKC 2004.
>
> [7] Publicly Detectable Watermarking for Language Models, arxiv 2024.

---

> > ### Comment · Reviewer_4vVx · 2024-08-07
> >
> > Thanks for the detailed rebuttal.
> >
> > I wish to first focus on the definition of "non-watermarked text" and "false positives".
> >
> > Support author A write an announcement of length 1000 with watermark. Then attacker B add/delete 10 words "not" to revert the meaning of the announcement. Is the changed announcement "watermarked text" or "non-watermarked text"? If watermark is detected, is it "false positive" or "true positive"?
> >
> > I would consider the changed announcement as "watermarked text", as 99% of the text is the same as original watermarked text, even though semantic is changed.
> >
> > Therefore, my previous comment is not `emphasizing the possibility of false positives in watermark detection`. Actually I didn't mention the word "false positive" at all and I regards the above example as "true positive".

---

> > > ### Author Response · Authors · 2024-08-07
> > >
> > > Thanks for your clarification. **What you mentioned highlights a significant flaw in prior watermarking schemes**: the potential for attackers to exploit the robustness of watermarks to achieve spoofing attacks, a vulnerability newly identified in a recent work [1] pointed out by Reviewer ggNL and in our work. **This issue is exactly what our scheme aims to address.** Our proposed method can differentiate between five cases, including whether the content is originally from the target LLM or has been modified.
> > >
> > > In particular, in the case you mentioned, other watermark schemes would identify the text as watermarked. In contrast, our detection first verifies the validity of the signature. In this instance, the signature would be invalid due to the perturbations, so we move to the next-level detection, i.e., checking if the text aligns with the key sequence. As reported in Table 3, the alignment with the key sequence is designed to be robust to perturbations. Thus, we can still detect the presence of the watermark in the text. **Ultimately, the combination of an invalid signature and good alignment with the key sequence indicates that the text originated from the target watermark but has been modified by others.**
> > >
> > >
> > > Thank you for your prompt response and the effort you put into reviewing our work. Please let us know if you have any further concerns.
> > >
> > >
> > >
> > > ---
> > > [1] Attacking LLM Watermarks by Exploiting Their Strengths, Pang et al. arXiv 2402.16187

---

> ### Comment · Reviewer_4vVx · 2024-08-07
>
> I still have doubt on the significance of spoofing attack. It slightly modify a watermarked content into another watermarked content, and I should expect the new content still be detected instead of not detected.
>
> If an author want to prove to others about: 1. whether LLM is used for generating the content, 2. whether content has been modified by others, can they simply use 1. watermark during generation 2. traditional digital signature to sign the generated content? It seems watermark detection can answer first question for reliable LLM detection and traditional digital signature can answer second question.
>
> > Ultimately, the combination of an invalid signature and good alignment with the key sequence indicates that the text originated from the target watermark but has been modified by others.
>
> The above scheme can do the same thing. The positive detection result and invalid signature indicates that the text originated from the target watermark but has been modified by others.

---

> ### Author Response · Authors · 2024-08-07
>
> Thank you for your timely response again! We understand the misunderstanding now.
>
> **The spoofing attack described in our work is NOT as you proposed above or described in the summary**, i.e., `an attacker wants to prove the proposition like "The person holding this watermark private key used an LLM to write this text A." where text A is constructed by the attacker. `
> Instead, the attack scenario is: when model owners deploy watermarks in their models, someone could forge the watermark in content that is not generated by the models or manipulate the content without removing the watermark. If the content is bad and the watermark's existence is detected, people would believe this was generated by the victim model. Then, the model owner could be blamed for not achieving safe deployment. We will clarify our threat model in the revision.
>
> Our work aims to help LLM model owners deploy a reliable watermark that cannot be spoofed by attackers.** The application scenario would be, for example, OpenAI deploying a watermark in ChatGPT, and ensuring that when someone claims content is generated by ChatGPT, it is indeed fully generated by it and not crafted by others or modified to twist its meaning. **This is different from someone wanting to demonstrate that they used an LLM to generate the entire content.**
>
> **To achieve our objective, your proposed method is invalid**, i.e., `they simply use 1. watermark during generation 2. traditional digital signature to sign the generated content.` The reason is that if the model first embeds existing watermarks and attaches the signature after generation, the output from the model would be the response to the prompt and a string of meaningless signatures. Thus, users would simply discard the signature when they copy and paste the response somewhere. As a result, no one can test if the content is truly from ChatGPT due to the missing signature.
>
> Therefore, our objective is to make the signature self-contained in the generated text. This way, if some generated text is given, OpenAI can extract the signature and verify if it is from ChatGPT. **Our work provides a reliable detection scheme for model owners, allowing them to protect their own interests by preventing spoofing attacks where attackers may mislead the attribution of bad content to the victim model.**

---

> > ### Author Response · Authors · 2024-08-12
> >
> > We appreciate your thoughtful review. As the rebuttal deadline approaches, we kindly ask if our responses have sufficiently addressed your concerns. Should you require further clarification, we are prepared to provide additional information.
> >
> > Sincerely,
> >
> > Authors

---

### Official Review · Reviewer_ggNL · 2024-07-13

**Soundness:** 2
**Presentation:** 1
**Contribution:** 1
**Rating:** 3
**Confidence:** 4

**Summary:**

The submission proposes a spoofing attack on LLM watermarks and a new bi-level scheme meant to protect against spoofing by distinguishing five possible scenarios. The scheme is based on signature bits for integrity checks and rank-based sampling on top of a Kuditipudi-like random key sequence.

**Strengths:**

- The paper takes a somewhat original approach compared to most contemporary methods.
- On a high-level, the problem of preventing spoofing is well-motivated and important for the community.

**Weaknesses:**

- Weak experimental results, bringing the practical value of the defense into question:
  - The provided quality evaluation, despite its limitations (see below), clearly shows an order-of-magnitude increase in perplexity which strongly suggests that produced text are of impractically bad quality; there is no evaluation that would test this. This is the most important weakness in my opinion.
- Limited experimental evaluation, in ways that make it hard to evaluate the merit:
  - Text quality is measured only as PPL of Llama-13B and only on one small 1.3B model; there is no qualitative evaluation of text quality so the negative effect on text quality can't be well understood.
  - Only Unigram and SLS are considered as baselines, while self-hash and other variants of the KGW scheme are generally considered more promising, esp. from the perspective of spoofing.
  - Watermark removal is evaluated only as 10% editing attack which ruins text quality, no paraphrasing attack is evaluated.
- Bigger framing issues around Table 2 and the attack:
  - The framing of Table 2 seems inappropriate. "Knowing the secret key" is not a spoofing attack but simply an application of the watermark, this seems to be introduced as a way to suggest that symmetric schemes are flawed by design, which is not necessarily true in cases where there is no detector access.
  - The attack is framed as a "novel advanced spoofing attack" while it is (1) in the opinion of this reviewer a direct result of scheme robustness and very limited in scope and thus hardly advanced (2) more importantly, already proposed in a different form in prior work [1] which was not cited, making this an overclaim. To elaborate on (1), for example, [7, 9] would be able to produce a detailed watermarked response to a harmful query such as "Teach me how to steal someone's identity" while there is no way to produce such a response by a few token modifications of a non-harmful response.
   - This attack type is used as a key motivation, setting aside the true spoofing attacks from [7,9], which are much more relevant. This is evident in claims such as "anti-spoofing requires perturbation-sensitivity". Further, the robustness of Bileve to such approaches based on learnability is claimed but not substantiated.
- Poor writing: The paper is often quite hard to read and understand. On top of that there is a very large amount of typos. I advise the authors to work on improving the writing for the next version. Here is a list of some examples that I found, in hopes this helps.
  - "Symmetric characteristic" and "learnability" in Introduction are unclear without being defined
  - Paper keywords typo: "provence"
  - L50: unforgettable
  - L285 L325 L50: tempering / temper-evident
  - Table 2: model'
  - L87: simply
  - Algo1: $h$ is undefined, although $H$ (a different symbol) is defined outside in the main text
  - L211: "associate"
  - L283: resulted
  - L284: "the source are"
  - L284: the failure verification
  - L308: "tokens also"
  - L311: "return"
  - L312: "the rest segments"
  - L314: "shows"
  - L315: "t0"
  - L316: "cause"
  - L327: "limitaition"
  - L456: "neucles"

[1] Attacking LLM Watermarks by Exploiting Their Strengths, Pang et al. arXiv 2402.16187

**Questions:**

- Can the authors provide evidence of practical text quality of Bileve texts?
- Can the authors include the missing experiments discussed above?

**Limitations:**

The authors include a discussion of limitations and societal impact in Section 6.

---

> ### Author Rebuttal · Authors · 2024-08-04
>
> Thanks for your input. We address the weaknesses (**W**) and questions (**Q**) below.
>
> **W1: Experimental Results**
>
> It is important to clarify that the increase in perplexity observed is **not an order-of-magnitude increase**. Additionally, **high perplexity does not necessarily indicate bad quality**. Perplexity is a measure of how well a probability model predicts a sample, and while it can be correlated with quality, it is not a direct measure of it.
> Moreover, perplexity does not always capture the nuanced aspects of human-readable text quality, such as relevance. This is why we are planning to incorporate other evaluation metrics, such as human evaluation, to provide a more comprehensive assessment of text quality. **It is noted that Bileve aims to establish a secure foundation against spoofing attacks. Compared to SLS, the perplexity is improved and shows potential for further enhancements as discussed in Section 6.**
>
> **W2: Experimental Evaluation**
>
> * More results have already been provided in Appendix E. To provide qualitative evaluation, we used zero-shot quality measurements with GPT-4 Turbo following [1], where a higher score indicates better quality. For the question-answering task using OPT-1.3B, the score of Unigram is 16±6.52 and 16±9.62. We would like to extend this evaluation of all tasks and models in our work in revision.
>
> * Unigram is a variant of KGW and retains the key merits of KGW. Its vulnerabilities are inherited from the robustness of KGW. As long as other variants like the self-hash of KGW are also robust against perturbation (as discussed in the KGW paper), they suffer from the same vulnerability. Could you provide evidence that self-hash is more promising in terms of defeating spoofing?
>
> * The objective of evaluation on watermark removal is to show that even at 10% editing, a stronger attack than <10% editing, the detectability is still well preserved. Therefore, <10% editing would not change the conclusion obtained from Table 3. Additionally, a semantic manipulation attack, a variant of paraphrasing attacks, is evaluated and the results are provided in Table 4.
>
> **W3: Framing Issue**
>
> * **We did not claim that "Knowing the secret key" is an attack; instead, it is an attacker’s capability**. We did suggest that symmetric schemes fall short in the case of using watermarks for transparent detection (instead of black-box API) or regulation.
> If "no detector access" refers to only providing a black-box API, then these watermarks cannot advance the societal goals of ensuring safe, transparent, and accountable LLM regulation as Bileve does. If there is no access at all, then it is unclear about the objective of deploying watermarks.
>
> * This spoofing attack does not aim to produce responses similar to jailbreaking (i.e., your example). Instead, if it can generate harmful content and cause damage to the model owner's reputation, it is considered a successful attack, and the consequence is also discussed in [2]. The term "advanced" is used to emphasize that attackers can achieve spoofing attacks with minimal capabilities.
> Also, thanks for pointing out this preprint work [1]. We acknowledge that it also proposes a method to achieve spoofing attacks, but our methods are different. Moreover, their work further highlights the vulnerability of spoofing attacks and underscores the need for effective defenses. While they suggest compromising watermark robustness to mitigate spoofing attacks, our work demonstrates how to achieve effective mitigation without sacrificing robustness. We will include a detailed discussion in the revision.
>
> * The claim of unlearnability in our work is based on cryptographic security principles, particularly those of digital signatures. This ensures that the watermark cannot be spoofed or tampered with [3], thereby maintaining the integrity and robustness of the system.
>
> **W4: Writing**
>
> * We clearly state in L30 that the explanations of `symmetric characteristic` and `learnability` are detailed in sec 2.3.
> * Thanks for your thorough reading and pointing out typos. We will fix them in the revision.
>
> **Q1: Practical Text Quality**
>
> In addition to examples in response to Reviewer 6UZB, we provide examples of LFQA tasks using LLaMa-7B, which show that **the higher PPL does not indicate impractically bad quality**.
>
> Ex1- Prompt:
> >Q: What does a Mayor even do?
>
> Unigram:
> >Uhhhhhhhhhhhhhhhhhhhhhhh...
>
> Bileve:
> >Most of the problems being experienced by our City are a result of bad planning, decisions, and practices of the City Council. Unfortunately, the City Council receives the majority of adulation for what\u2019s going on in the City.\nThe mayor is the City Manager...
>
> Ex2- Prompt:
> >Q: Mandatory arbitration
>
> Unigram:
> >I am sorry for this but I am out of answers. I will ask others for a solution. Thank you. Please ask more questions later on. If you wait 2 hours I will be back...
>
> Bileve:
> >Mandatory arbitration is a means for eliminating affected commerce and eliminating employees' rights to sue as private citizens. Unions do not like the term...
>
> Ex3- Prompt:
> >Q: when does a case need jurors?
>
> Unigram:
> >a court can order a jury as a court order. If a party asks for a jury. If a party appeals a court. If a party files a law suit (which must be done before a court can be held)...
>
> Bileve:
> >Tuesday at all times and Thursdays at 9:00 am. If you are qualified, you may be called for a case or cases may be filled from qualified jurors already on the list...
>
>
> We would like to include additional metrics, like the one in **W2** and human scoring, to evaluate quality more comprehensively in the revision.
>
> **Q2: Additional Experiments**
>
> When the editing ratio is 5% (measured by TPR on OPT-1.3B):
> |          | OpenGen  | LFQA    |
> |----------|----------|---------|
> | Unigram  | 0.993   | 0.998  |
> | Bileve   | 0.999  | 0.999  |
>
> ---
> [2] Watermark Stealing in Large Language Models, ICML 2024.
>
> [3] Publicly Detectable Watermarking for Language Models, arXiv 2024.

---

> > ### Author Response · Authors · 2024-08-12
> >
> > As the discussion period is about to conclude, we would appreciate it if you could review our above response and let us know if our rebuttal addresses your concerns. Thanks!

---

> > > ### Comment · Reviewer_ggNL · 2024-08-12
> > >
> > > I thank the authors for their rebuttal. I maintain most of my concerns; to comment on some:
> > > - The authors provide PPL as the metric used to measure text quality. Saying that this metric is sometimes unreliable and providing anecdotal evidence is insufficient. If the authors agree that the metric is flawed the evaluation should be repeated with a better one.
> > > - A paraphrasing attack, as commonly studied in recent work, is significantly stronger than the proposed semantic attack. Thus this evaluation is still missing.
> > > - The claim that Self-hash is more robust to spoofing than Unigram comes directly from the scheme properties and can be found also in the original Unigram paper (see bottom of Sec 2). Authors failing to acknowledge this is for me a significant oversight.
> > >
> > > I encourage the authors to remedy the above as well as improve the poor writing as promised in the next revision of the paper.

---

> > > > ### Author Response · Authors · 2024-08-14
> > > >
> > > > It is unfortunate that we received your feedback so late, which has limited our ability to address all your concerns thoroughly before the deadline. Nevertheless, we want to clarify key points to assist in your evaluation of our work.
> > > >
> > > > **Generation Quality**
> > > >
> > > > We include PPL as it is a common metric used to evaluate generation quality. However, we mentioned other metrics mainly because **your assertion that our method yields “impractically bad quality” based solely on PPL is incorrect.**  Our method’s perplexity is indeed higher than Unigram, but that does not directly equate to impractically bad quality (and we also discussed how to improve it in the limitation section). Therefore, by learning from recent literature, we acknowledged that additional metrics can be provided for comprehensive evaluation.  We included it in our initial rebuttal and will add it to the final version.
> > > >
> > > > **Paraphrasing Attack**
> > > >
> > > > As we explained in the initial rebuttal, `a semantic manipulation attack, a variant of paraphrasing attacks, is evaluated and the results are provided in Table 4.` **Our work is designed to ensure that model owners can deploy a reliable watermark that is not easily misattributed.** Evaluating paraphrasing attacks comprehensively is valuable, but **it does not alter the fundamental conclusion that our method is robust against various spoofing attacks**, since such attacks would break the watermark signature, leading to verification failures.
> > > >
> > > > **Self-hash**
> > > >
> > > > Our paper includes an overview of spoofing attacks in Table 2. We did not include self-hash in our comparison because it shares similar vulnerabilities with Unigram, and **has been successfully spoofed as demonstrated in the reference [9] in Table 2**. That is why we are puzzled by your claim that self-hash is promising for defeating spoofing attacks.
> > > >
> > > >
> > > >
> > > > ---
> > > > [9] Watermark Stealing in Large Language Models, ICML 2024.

---

### Comment · Area_Chair_EFGe · 2024-08-10
**Please engage in discussion**

Dear Reviewers,

The authors have provided responses to the reviews. If you have not already done so, please take a look at the responses (and other reviews) and engage in a discussion.

Thanks for your service to NeurIPS 2024.

Best,
AC

---

### Decision · Program_Chairs · 2024-09-25

**Decision:**

Accept (poster)

**Comment:**

This work introduces a watermarking method for LLMs that is resistant to spoofing attacks, yet can be robustly recovered. The method is based on digital signatures and improves upon the existing SLS method. Reviewers appreciated that the paper tackles the important problem of watermark spoofing in a novel way but had concerns about text generation quality and evaluation. There was a good discussion during the rebuttal period which clarified the use-case and concerns about text generation quality due to the use of perplexity as a metric. Some reviewers had remaining concerns about the robustness evaluation and presentation.

On balance, the AC thinks that the novel approach, promising results and timeliness of the work outweighs the remaining concerns related to text generation quality, robustness evaluation and presentation. The AC hence recommends that this paper be accepted. That said, the authors should include the revisions as promised in the rebuttal, provide additional context on the perplexity evaluation (and possibly alternate measures of quality like benchmark task performance) in addition to addressing reviewer concerns regarding presentation of the work.